# Multisensor Integrated Platform Based on MEMS Charge Variation Sensing Technology for Biopotential Acquisition

**DOI:** 10.3390/s24051554

**Published:** 2024-02-28

**Authors:** Fernanda Irrera, Alessandro Gumiero, Alessandro Zampogna, Federico Boscari, Angelo Avogaro, Michele Antonio Gazzanti Pugliese di Cotrone, Martina Patera, Luigi Della Torre, Nicola Picozzi, Antonio Suppa

**Affiliations:** 1Department of Information Engineering, Electronics and Telecommunications, Sapienza University of Rome, 00185 Rome, Italy; gazzantipugliesedicotrone.1744018@studenti.uniroma1.it; 2STMicroelectronics Agrate, 20864 Agrate Brianza, Italy; alessandro.gumiero@st.com (A.G.); luigi.dellatorre@st.com (L.D.T.); nicola.picozzi@st.com (N.P.); 3Department of Human Neurosciences, Sapienza University of Rome, 00185 Rome, Italy; alessandro.zampogna@uniroma1.it (A.Z.); martina.patera@uniroma1.it (M.P.); antonio.suppa@uniroma1.it (A.S.); 4Department of Medicine, University of Padua, 35122 Padua, Italy; federico.boscari@unipd.it (F.B.); angelo.avogaro@unipd.it (A.A.); 5IRCCS Neuromed, 86077 Pozzilli, Italy

**Keywords:** wearable sensors, long-term biopotential recording, MEMS technology, charge variation sensors, low power consumption

## Abstract

We propose a new methodology for long-term biopotential recording based on an MEMS multisensor integrated platform featuring a commercial electrostatic charge-transfer sensor. This family of sensors was originally intended for presence tracking in the automotive industry, so the existing setup was engineered for the acquisition of electrocardiograms, electroencephalograms, electrooculograms, and electromyography, designing a dedicated front-end and writing proper firmware for the specific application. Systematic tests on controls and nocturnal acquisitions from patients in a domestic environment will be discussed in detail. The excellent results indicate that this technology can provide a low-power, unexplored solution to biopotential acquisition. The technological breakthrough is in that it enables adding this type of functionality to existing MEMS boards at near-zero additional power consumption. For these reasons, it opens up additional possibilities for wearable sensors and strengthens the role of MEMS technology in medical wearables for the long-term synchronous acquisition of a wide range of signals.

## 1. Introduction

In the near future, wearable technologies will pervade the medical area [1]. Ubiquitous monitoring and remote supervision make wearables extremely appealing for clinicians, making instrumental exams accessible in domestic environments. Among others, this is the case of electroencephalograms (EEGs), electrocardiograms (ECGs), and electromyography (EMG), on which research is spending great efforts to provide wearable variants [2,3,4].

EEGs are usually performed by trained personnel in ambulatory settings, recording several channels by placing numerous electrodes across the scalp, with cables connecting to the instrumentation, thus resulting in a lengthy and uncomfortable setup procedure. Although this analysis provides fundamental and exhaustive insights into a variety of disorders, this is not always the best way to obtain the desired information from the EEG. Indeed, in specific circumstances or for certain diseases, it would be preferred to monitor even just one EEG channel, but continuously for several days, in a domestic environment and free-living conditions, rather than several channels for a short time in hospital. For example, this is the case for re-enabling communication through brain–computer interface approaches in patients suffering from advanced stages of paralyzing disorders, such as amyotrophic lateral sclerosis (ALS) [5]. Today, portable EEG devices are also available, but their bulkiness, obtrusive setup, and limited battery life underscore the need for a wearable technology offering both prolonged and comfortable monitoring, which is more crucial than obtaining multiple channels. Moreover, by providing real-time, long-term, and free-living dynamic information, new non-invasive wearable solutions for EEG monitoring would support the differential diagnosis of pathological conditions characterized by transitory changes in focal brain activity, including epileptic and non-epileptic disorders, such as the loss of consciousness (i.e., cardiovascular syncope) and parasomnias (i.e., REM behavior disorder) [6,7]. As a testament to the importance of portability, in the last few years, several portable EEG devices have appeared on the market [8,9,10,11,12,13,14,15,16], including in-ear EEG sensors [17,18]. However, none of these devices has successfully addressed the dual challenge of providing a battery lifespan longer than 12 h and ensuring sustained comfort for prolonged in-home applications.

Regarding ECGs, it is noteworthy that that this analysis encounters similar challenges to EEGs in terms of the test environment, comfort, and duration. Indeed, comprehensive monitoring over several days is essential for diagnosing paroxysmal arrhythmias, evaluating potential arrhythmic symptoms, and understanding syncope mechanisms [19]. ECG Holter monitors, which usually acquire information for up to 48 h, often fail to identify heart rhythm disturbances due to limited monitoring duration [20]. The likelihood of identifying an arrhythmia like atrial fibrillation notably increases with extended monitoring, carrying substantial therapeutic implications [21]. Indeed, to prolong the monitoring duration of cardiac activity, a “loop recorder” is occasionally adopted. However, this solution demands the invasive subcutaneous implantation of a device, thus limiting its widespread application. Unfortunately, most commercially available wearable systems are limited either in their recording sessions (the recording automatically stops after 30–300 s and has to be restarted) or in their battery life, preventing continuous monitoring for durations exceeding 24 h [22].

When considering EMG, its importance is evident in cases where monitoring muscle activity and distinguishing between involuntary and voluntary muscle contractions are crucial, such as in specific movement disorders associated with neurological diseases. Surface EMG (sEMG) uses patch electrodes and is not invasive, offering greater comfort and tolerance for patients compared to needle-based EMG. Moreover, the capability for multichannel operation allows for the acquisition of signals from multiple muscles simultaneously [23,24]. Traditionally, sEMG is performed within healthcare facilities using wired equipment, presenting several challenges. Firstly, these tests are usually short-lasting and conducted in non-domestic environments; thus, they are not truly representative of clinical conditions. Secondly, the presence of wires imposes spatial constraints on patients, restricting their range of movement. Furthermore, the wired connection to the electrical grid introduces noise at 50 Hz, which coincides with a crucial segment of the EMG spectrum in many tests. To overcome these limitations, in recent years, wireless wearable sEMG devices have been proposed for use in a very wide range of applications, and some products are already on the market (see, for example, [25,26,27,28]). However, for wearable sEMG systems, the applications are also limited by their short battery life. Indeed, a comparison of products on the market shows that the lowest power consumption is some tens of mW, resulting in battery lifespans of less than 8 h [29,30,31,32]. Very recently, an s-EMG system has been presented in the literature that supports high sampling and lower power consumption for measurement [33]. This innovative approach optimizes each part of the EMG acquisition circuit and combines an MCU with BLE for transmission to a smartphone. By employing a smart signal management strategy, the authors claim an extended battery life longer than that of commercial devices.

In summary, despite the clinical importance of recording data over extended periods, current commercial wearable ECG, EEG, and EMG devices show limited battery life, typically lasting 24 h for ECG, 12 h for EEG, and even less for EMG. These energy limitations are in conflict with the envisioned days or weeks of monitoring and do not allow space for future developments. Indeed, the existing constraints in processing power and device battery life not only preclude the integration of sophisticated machine learning algorithms [34] but also make it unfeasible to directly embed artificial intelligence on the boards. In addition, the energetic constraints limit the capacity to acquire a broader spectrum of information, including synchronously recorded biopotential, along with other physical or physiological parameters, using a single device—a potentially valuable diagnostic approach in many cases. Long-term monitoring, polygraphic recording, and embedding smart algorithms at the same time would represent a breakthrough for many applications in medicine. From such an overview, it is clear that new strategies and technologies are necessary to reduce the power consumption of wearable sensing systems and improve their functionality and versatility.

In this context, we propose a commercial MEMS multisensor integrated system featuring a charge-transfer sensor (QVAR), designed for long-term recording of biopotential and inertial data. Originally developed for presence tracking in the automotive industry, we innovatively engineered the QVAR sensor to adapt the existing setup to biopotential acquisition, designing a dedicated front-end and writing the proper firmware. This technology provides an incredibly low-power, unexplored solution to biopotential acquisition, and it also enables adding this type of functionality to any existing MEMS board at near-zero additional power consumption (<20 μA per channel). For these reasons, QVAR sensors open up additional possibilities for wearable sensors and strengthen the role of MEMS technology in medical wearable applications for the long-term synchronous acquisition of biopotential and inertial signals. In a previous paper [35], we presented preliminary results related to the acquisition of ECGs and electrooculography (EOG) with QVAR sensors. In this paper, we propose the acquisition of EEG channels, with specific reference to alpha and beta waves, and a systematic study of sEMG applied to different muscles. Finally, we combine the aforementioned functionalities and present two QVAR-based wearable systems applied to different case studies. In particular, in one case, we combined the sensing of EEGs and ECGs for the domestic monitoring of vital signs in hypoglycemia while, in the other case, we combined sensing of sEMG, ECGs, and EOG for REM/NREM sleep screening. To the best of our knowledge, this is the first time that electrostatic sensors based on electric charge transfer have been used in practical healthcare applications.

## 2. Materials: The QVAR Sensor

### 2.1. Operational Principles

The focus of this paragraph is the QVAR sensor, an electrostatic charge variation sensor. It is commonly incorporated into MEMS chips that are manufactured by STMicroelectronics (Agrate Brianza, Italy).

Although the QVAR sensor was originally designed as an automotive sensor, it appears to be well suited for use in the field of wearable technology as well. The only difference is that, in the medical context, electrodes are connected to its inputs, and the system is battery-powered.

Usually, the QVAR analog front-end (AFE) is combined with a finite-state machine (FSM) and a machine learning core (MLC). This combination enables outstanding edge-processing capabilities. The QVAR evaluation board is depicted in Figure 1a. The QVAR sensor, with its differential input, is capable of detecting variations in electrical potential induced over its two electrodes. This can occur directly, through a charge-transfer mechanism, or indirectly, through an insulator using electrostatic induction. It is used in various applications, such as presence detection, touch sensing, and user interface creation.

The QVAR sensor is available in a small, thin, plastic land grid array (LGA) package, and it is guaranteed to operate over an extended temperature range, from −40 °C to +85 °C.

This functionality enables the detection of changes in biopotential with the addiction of minimal electronics required to condition the inputs. The conditioning process is essential to polarize the electrodes and amplify the overall gain of the acquisition chain based on the signal being analyzed. Electrode polarization helps prevent saturation that may be caused by the body’s coupling with external potential originating from electromagnetic disturbances.

Eventually, the QVAR sensor allows for a very simple, low-cost, and low-power reading of vital signs. QVAR sensors offer high performance and ultralow power for long battery life. The AFE channel has low noise, a high common-mode rejection ratio (CMRR), and programmable gain and input impedance.

Figure 1b shows the block diagram of the QVAR biosensing channels is. The blocks are as follows:Electrodes: The electrodes are always necessary to read the signals. Usually, they are made of copper, silver, tin, or gold and can change in dimensions. They can be directly coupled to the skin, as with the wet electrodes, or isolated from the skin using electrostatic induction. Apart from the case of specific acquisitions from the scalp, where gold cup electrodes in conjunction with conductive paste were used, we always used wet silver–silver chloride (Ag/AgCl) patch electrodes directly on the skin, which guarantee good and stable electrical and electrochemical properties and excellent signal acquisition performance. It is important to reduce the series resistance introduced by the electrodes to a minimum, but the high input impedance of the QVAR sensor helps in this.AFE: This is an analog front-end, which performs the conditioning and the amplification. External amplification is not always necessary.ADC: This is a 12-bit analog-to-digital converter.Digital processing unit: This is composed of a finite-state machine and a machine learning core.

Figure 1c provides a detailed analysis of the entire QVAR sensor chip. The main ADC chain is shared between two primary input stages: the MEMS section, which is dedicated to inertial measurements, and the differential analog input, which is dedicated to biosignal measurements. Both inputs are processed by the analog front-end and then sent to the digital logic component for further processing before being transmitted via digital interfaces.

The digital logic section allows for programmable sampling averaging of the acquired signal, as well as digital compensation to remove any artifacts prior to filtering the signal. The acquired samples can be stored in a FIFO memory or accessed through dedicated output registers.

### 2.2. Electrical Features

As previously mentioned, QVAR sensors are typically integrated into pre-existing MEMS devices that serve various functions. This integration is highly advantageous in many cases. For instance, it enables the simultaneous acquisition of bioelectrical signals and motion information, which is crucial in correcting artifacts, without the need for complex electronics or increased power consumption.

In Table 1 and Table 2, typical values of the principal electrical parameters and characteristics of the QVAR sensor, respectively, are reported at VDD = 1.8 V and room temperature.

### 2.3. Potentiality and Limits of the QVAR Sensor

Currently, QVAR sensors provide an optimal and easily integrated solution for simultaneously performing different readings of biopotential and inertial parameters. This is achieved while keeping the energy consumption and cost of the final solution almost unchanged.

Unlike signals such as ECGs, EMG, and EOG, for the acquisition of EEG signals, the actual QVAR sensor requires the use of external amplification to detect those weak electrical signals. Increasing the gain of the internal chain and improving the performance in terms of input noise and CMRR would easily solve this need. It would also provide greater immunity to external disturbances, potentially allowing the system to operate without a third reference electrode to polarize the human body during measurement.

A new version of the QVAR sensor, called vAFE, is currently being developed by STMicroelectronics, which, in addition to the abovementioned requirements, can also guarantee greater input impedance. This allows for a better transfer of signals from the electrodes to the acquisition chain, making it possible to use dry electrodes for measurements. With these improvements, vAFE sensors can potentially acquire any type of biopotential without the need for external electronics and reference electrodes. From here, this new product will be used in this kind of application. The vAFE sensor is specifically designed for biosignal detection, and the overall performances of the system are improving.

## 3. Methods: Biopotential Acquisition by QVAR Sensors

### 3.1. The Starting Point: Method of ECG Acquisition

In Ref. [35], we have already demonstrated methods for the correct acquisition of ECGs and EOG by QVAR sensors. Those methods are briefly recalled in this paragraph, since they will be used in combination with other functionalities of the QVAR sensor for the realization of the two systems described in the next section.

To record the ECG signal with the QVAR sensor, we used a single-lead ECG setup and determined the heart frequency by detecting the R-peaks and the evaluation of RR intervals. Specifically, a lead D1 configuration, placing positive and negative electrodes on the left and right upper extremities, respectively, was adopted. Wet Ag/AgCl electrodes were positioned in RA-LA configuration within the Eindhoven triangle, as illustrated in Figure 2a [36], and the trace was recorded for an extended duration (i.e., tens of seconds). The signal underwent filtration in the 1–35 Hz band. Additional details on the technique of ECG recording with QVAR sensors can be found in [35]. Notably, typical features such as the P wave, QRS complex, and T wave were distinctly recognizable [37], as highlighted in Figure 2b. The extraction of RR intervals between consecutive oscillations allows for the calculation of the hearth rate (HR). Another clinically significant metric that can be obtained from this trace is the QT length, along with its normalized value (i.e., QT_C_ = QT/RR^1/2^). In multichannel ECG acquisitions, variations in electrode location change the morphology of the ECG waveform. However, the R-peak period and, consequently, the length of the RR intervals remain independent of the chosen derivation. Accordingly, the HR measurement derived from the RR intervals is derivation-independent [38].

To prove the robustness of our sensing technique with respect to the derivation, we conducted a simultaneous acquisition using both the QVAR sensor and a gold-standard system (the certified Micromed system), positioning the electrodes in distinct configurations. Referring to Figure 2a, we adopted the RA-LA configuration for the QVAR system and the RA-LL configuration for the gold standard. In Figure 2b, both traces are presented, with dashed vertical lines added for clarity to highlight the R peaks. Notably, the R peaks from the QVAR system were perfectly synchronized with those from the gold standard, despite the adoption of different derivations. The differences in waveform morphology, as shown in the figure, including the difference relative to the T wave, were due to the different derivation. These differences are meaningless from a clinical viewpoint and do not affect the diagnostic utility of the system for monitoring cardiac rhythm. Based on the use of a single-lead ECG, our approach would be suitable for the basic monitoring of heart rhythms, enabling the detection of arrhythmias, such as atrial fibrillation, and the examination of pathological HR variability (HRV) [39].

### 3.2. The Starting Point: Method of EOG Acquisition

To record the electrooculogram (EOG) signal with the QVAR sensor, in Ref. [35] we simultaneously acquired a single-channel EEG using both QVAR and a gold-standard system (Micromed, Rome, Italy). Wet Ag/AgCl patch electrodes were placed on the frontal lobes at locations Fp1–Fp2 following the International 10–20 System Standard [40]. Indeed, the EOG derives from an artefact in the frontal channels of the EEG [41], due to eye blinking and movements. To ensure comparability, the electrodes of the gold standard (Micromed) and the QVAR sensor were positioned as closely as possible. During this setup, the subject was instructed to keep their eyes closed. The raw traces from a short interval are shown in Figure 3; the blue line refers to the QVAR recording, while the red trace refers to the gold-standard recording. Notably, the two traces overlap significantly. The peak at 61 s, recorded by both systems, corresponds to a blinking event.

Analysis of the EOG can be useful in specific medical areas [42,43]. One of the main applications of EOG is sleep staging [44]. Indeed, the so-called “rapid eye movement” gives the REM sleep phase its name. This functionality of the QVAR sensor will be used in the next section, in combination with ECG and sEMG, towards REM–non-REM sleep staging.

## 4. Results and Applications

### 4.1. Acquisition of α-Wave and β-Wave EEGs

We acquired α-wave and β-wave EEGs using QVAR sensors and compared the results with those of a gold standard (the Cyton BCI). To ensure comparability, the electrodes of the Cyton BCI and QVAR sensors were always positioned as closely as possible. Placing gold cup electrodes with a conductive paste at occipital sites O1 and O2 of the International 10–20 System Standard [40], and then filtering within the 8–13 Hz band, maximized the specific contribution of α waves. Figure 4 displays the normalized filtered traces of QVAR and the gold standard during a test where the subject remained relaxed, with eyes closed. The typical fusiform pattern is distinctly visible, with approximately ten peaks in each fuse. Once again, the agreement of the two traces is excellent.

To maximize detection of the β waves, we placed the wet Ag/AgCl patch electrodes on the frontal lobes at locations Fp1–Fp2 and filtered in the 1–3 Hz band [40]. Figure 5 displays the normalized filtered traces of QVAR and the gold standard during a test where the subject remained relaxed, with eyes closed. The peaks are artifacts of head movements, blinking, and eye rotation. The comparison reveals an excellent performance of the QVAR sensor in detecting the β-wave EEG channel (and distinguishing the eye activity).

### 4.2. Acquisition of sEMG

Wearable sEMG has found widespread commercial applications, ranging from gaming to rehabilitation medicine, analysis of motion, analysis of muscle fatigue, and prosthesis control. Less popular applications include drowsiness control in car drivers [45], the study of hand gestures for reproducing sign language [46], monitoring involuntary activity of the jaw muscles [47], and analyzing facial expressions [48]. In the aforementioned applications, the sEMG signal amplitude depends on the specific muscle under study and the type of activity, but it is rarely higher than few millivolts. Conversely, the power spectrum of a typical raw EMG signal has consistent frequency components, irrespective of the muscle being examined or the type of physical connection (needles or patch electrodes). The band of interest consistently falls between 3 and 500 Hz, with the primary content concentrated between 10 and 250 Hz. Conventionally, this analysis is performed during outpatient visits by means of wired research laboratory equipment. It involves three electrodes properly placed on the skin, with positive and negative electrodes positioned over the target muscle and a third, reference electrode placed at a distance from it.

In our experiments, we focused on two distinct muscles, namely, the flexor carpi radialis and the facial masseter, which are characterized by a very different conformation and activation intensity. The experimental protocol involved the subject performing the following simple sequence a few times: 5″ contraction followed by 5″ relaxation, starting from a relaxed condition. To validate the test, the exercises were conducted with the subject concurrently wearing the QVAR wearable device and the EMGClick by MiKroElectronika, a commercial wearable device established as the gold standard [49]. The primary objective of the test was to ensure that the QVAR device accurately detected muscle activation without generating false activations. This test is of the on/off type, without any intensity evaluation. The collected signals underwent the following processing steps: (1) HP filtering at 3 Hz to remove DC and movement artifacts, along with notch filtering at 50 Hz; (2) rectification; (3) smoothing with LP filtering at 10 Hz to maintain the envelope; (4) normalization. Each experiment was systematically repeated by inverting the position of the QVAR and EMGClick electrodes, in order to verify the independence of the results on electrode placement. In Figure 6, the wet Ag/AgCl patch electrodes positioned on the arm to test the activity of the flexor carpi radialis are depicted. The black-and-white electrodes belong to the QVAR device, while the others pertain to the gold standard, including the reference electrode on the wrist. In the same figure, two traces are displayed: the one collected by the QVAR device (top), and the one collected contemporarily by the EMGClick device (bottom). As one can see, the QVAR device precisely reports muscle activity, demonstrating exact synchronization with the gold standard within a fraction of second.

Regarding the acquisition on the masseter, as illustrated in Figure 7, patch electrodes on the face were employed to test activity of the masseter. Again, the black-and-white electrodes belong to the QVAR device, while the others belong to the gold standard, including the reference electrode on the neck. The displayed traces show a clear and synchronous recording of muscle activity by the QVAR (top trace) and EMGClick (bottom trace) devices.

As a final remark, it is noteworthy that that this is the first reported application of QVAR sensors for sEMG recording.

### 4.3. Case Study: Domestic Monitoring of Vital Signs in Hypoglycemia

In this paragraph, we describe a practical application in healthcare. The system described in this case study includes two QVAR sensors—one for the acquisition of ECGs and one for the acquisition of α-wave EEGs—and is intended for the domestic monitoring of biopotential before hypoglycemic crisis.

Diabetes is a metabolic disorder characterized by high blood glucose that affects around half a billion persons worldwide today, with an estimated yearly cost of 10% of global health expenditure. Type 1 diabetes (T1D) is due to insulin deficiency and is treated pharmacologically by insulin administration, which in any case cannot replicate the physiological secretion and, thus, leads to glucose oscillations. Hypoglycemia (HG) is a major threat in T1D patients and commonly occurs in clinical practice in approximately 90% of all patients who receive insulin [50]. In average, individuals with T1D experience about two episodes of symptomatic hypoglycemia per week. The prevalence of severe hypoglycemia, which necessitates assistance for recovery, is approximately 30–40% annually, with an incidence of 1.0–1.7 episodes per patient per year. The likelihood of experiencing this risk increases considerably with prolonged disease duration [51]. Symptoms of hypoglycemia can be mainly divided into two categories: neurogenic (autonomic), and neuroglycopenic. Neuroglycopenic symptoms are caused by brain glucose deprivation and include cognitive impairment (altered perception, poor concentration, slow speech, slow decision-making), behavioral changes (irritation, frustration), psychomotor abnormalities (weakness, incoordination), seizure/coma, and permanent neurological damage for prolonged severe hypoglycemia [52]. Neurogenic symptoms, on the other hand, are caused by the sympathoadrenal response and include adrenergic symptoms (palpitations, tremulousness, anxiety, arousal, skin pallor/flushing or blotchy rashes, tingling around the mouth/lips) and cholinergic symptoms (sweating, hunger, paresthesia) [52].

Nocturnal HG is particularly dangerous because symptoms are typically blurred by sleep, sometimes resulting in coma and even death. It is known that, during the transition to HG, in order to bring blood sugar levels back up to normal, the body elicits a hormonal response that, in turn, leads to a number of symptoms. These include heart rate variability, along with progressive degradation of cognitive functionality and cerebral activity.

To gain deeper insight into the evolution of vital signs during the transition to HG, we propose a wearable system based on QVAR sensors recording electrocardiograms and electroencephalograms during the night, for use in domestic environments. The strategy would be to correlate those biopotential readings with the level of glucose in the blood, as measured instantaneously by a glucose sensor. Some authors have identified a meaningful set of parameters to be extracted from ECG and EEG traces in the time and frequency domains, which have been observed to be altered before the onset of an HG episode. From ECG, they focused on the QT_c_ interval length, the RR tract, the HR and the HRV with its standard deviation normal-to-normal (SDNN) and its root-mean-square of successive differences (RMSSD) and, finally, the low/high-frequency ratio (LF–HF) of the HRV power spectrum [53,54,55,56,57,58,59,60,61]. Regarding the EEG, only a few authors have reported results starting from the power spectrum of α waves and calculating the centroid frequency (CF), along with the spectral rotational radius derived from spectral moments G0 and higher [62,63,64].

The hardware of our system consists of two boards, each integrating a QVAR sensor, a microcontroller, a battery, and microSD memory. The two boards are fixed on the chest: one sensor acquires ECGs with two electrodes in the RA-LA positions, while the other sensor acquires the α-wave EEG with two electrodes on occipital sites O1 and O2. The data are all stored on the microSD card. Preliminary verification of the correct operation of the QVAR system was performed on control subjects, by comparison with a gold standard (Cyton BCI, in this case). Table 3 summarizes the values of the parameters calculated from the ECGs and EEGs, averaged over 24-h acquisition on six control subjects; as one can see, the agreement is excellent, with a discrepancy lower than 1.5%.

In Figure 8, the RR trait duration extracted from the ECGs with our system (top) and the gold standard (bottom) are compared in the time and frequency domains. In the same figure, the α-wave ECGs recorded with our system (top) and the gold standard (bottom) are compared in the time and frequency domains.

After those preliminary tests, a T1D patient was studied overnight in a domestic environment. He was male, 63 years old, diagnosed at age 22, treated with an insulin pump, and had a glucose target value of 120 mg/dL (this relatively high value should not be surprising, since each diabetic patient has a specific glucose reference value and a specific hypoglycemia threshold). The patient always puts on a glucose sensor (GS) on an arm, recording his glucose value every minute, with his threshold for hypoglycemic alarm set at 80 mg/dL. The patient gave informed consent to be tested with our device. So, throughout the testing nights, he was wearing the QVAR system in addition to the gold standard and the GS. At a certain time (which hereafter will be referred to as t = 0), the glucose reached the critical threshold and the GS alarmed the patient, who woke up and immediately consumed sugar. In Figure 9 and Figure 10, we show the values (orange curves with symbols) of the ECG parameters (LF:HF, SDNN) and EEG parameter (CF), respectively, reported by the QVAR system before the alarm (t = 0). In the same figures, the glucose data recorded by the GM are also drawn as solid blue lines. A dashed line is also drawn to highlight the intervals where the ECG and EEG parameters exhibited a meaningful increase, in contrast with the intervals where their averages remained constant.

As one can see, the ECG underwent a slow and progressive variation when the glucose decreased below the value of 100 mg/dL, long before the alarm at t = 0 (approximately 90 min before), while the α-wave EEG exhibited a sudden variation only 15 min before the alarm.

At this stage, any conclusion from the clinical viewpoint would be meaningless, since it would require much further study and many more patients (beyond the scope of this paper); however, the technological conclusion is that our wearable system using QVAR sensors can efficiently monitor ECG and EEG channels simultaneously, allowing us to study vital signs before and during nocturnal HG crises in a domestic environment.

### 4.4. Case Study: Domestic Monitoring of Non-EEG Biopotential for REM/NREM Sleep Screening

In this paragraph, we describe another practical application in healthcare. The system described in this case study includes three QVAR sensors—one for the acquisition of ECGs, one for the acquisition of sEMG, and one for the acquisition of EOG—and is targeted to the domestic monitoring of sleep for REM/NREM screening.

Sleep can be disrupted by various disorders that are classified based on the stage of sleep during which they manifest (i.e., rapid eye movement (REM) and non-rapid eye movement (NREM)). These disorders exhibit distinct pathophysiological mechanisms and prognoses, and they require specific therapeutic interventions [65]. Accordingly, the accurate diagnosis and treatment of different sleep disorders crucially depends on the ability to properly differentiate between sleep phases. One notable example is REM sleep behavior disorder (RBD), a parasomnia occurring during the REM sleep stage, which is particularly significant for neurologists, psychiatrists, and geriatricians. Indeed, RBD is acknowledged as a precursor to neurodegenerative diseases such as Parkinson’s disease, dementia with Lewy bodies, and multiple system atrophy [66,67]. Manifesting as dream enactment behavior [66], patients with RBD engage in activities like screaming, speaking, falling, and mimicking dream motions, often resulting in injuries and violence. The diagnosis of RBD is usually challenging, necessitating a history of complex sleep behaviors confirmed during REM sleep. REM sleep should therefore be demonstrated with the contemporary absence of atonia by evidencing dream enactment behavior [68]. The gold standard for the diagnosis of sleep disorders is polysomnography (PSG), according to the American Association of Sleep Medicine (AASM)’s scoring guidelines. However, PSG is a cumbersome test available in a very limited number of medical centers, requiring specialized supervision and leading to prolonged waiting times. Indeed, difficulties in access to PSG contribute to undiagnosed and untreated sleep disorders, with significant consequences. Moreover, it has been demonstrated that the environment influences sleep quality [69], so that the PSG itself may be distorted by the altered emotional state of the hospitalized subject. Accordingly, the challenge with diagnosing RBD extends beyond the need for hospitalization to conduct PSG. Indeed, even after the test is performed, it may not be effective, since symptoms do not always appear, necessitating extended hospital stays for an accurate diagnosis. As a consequence, RBD still remains largely undiagnosed [70], like many other sleep-related disorders. Since RBD may precede specific neurodegenerative diseases by several years, recognizing it becomes crucial for identifying at-risk patients during the preclinical phase. The early identification of individuals at risk of neurodegenerative disorders would allow for the development and administration of novel neuroprotective drugs, potentially delaying or preventing the onset of specific diseases [71].

Due to all of these factors, the field of telemedicine is currently focusing on more widespread screening and home monitoring approaches for sleep disorders [69,72,73]. The aim is to support clinicians in their diagnostic procedures. Domestic screening facilities would play a crucial role in identifying at-risk individuals who urgently need hospitalization for diagnosis, thus optimizing the utilization of limited PSG resources and the expertise of trained personnel. This approach would reduce the waiting times from recognition to diagnosis and treatment of sleep disorders, ultimately increasing adherence to treatment, despite a growing number of sleep consults [74,75]. In the last few years, portable solutions and systems for domestic screening, detecting, and/or remote monitoring of sleep have been proposed. These can be categorized into two main categories: those using EEG, and those that do not.

As mentioned earlier, PSG provides the most accurate and comprehensive insights into sleep patterns. It identifies the REM stage by detecting epochs characterized by low-amplitude, mixed-frequency EEG, low chin/masseter EMG tone, and rapid eye movements [76]. On the opposite side, we can find examples of far less accurate wearable devices, such as commercial sleep trackers (CSTs), which do not use EEGs and focus on fitness rather than on healthcare. These devices rely on measuring the heart rate (HR), HR variability (HRV), and movements using photoplethysmography (PPG) and inertial sensors. While they can distinguish between the wake and sleep phases [77,78,79,80,81,82,83,84,85], they are totally unaffordable for the comprehensive analysis of sleep [86,87,88]. With this information, it is clear that HR, HRV, and movements are not fully representative of the REM phase. Using artificial intelligence (AI) can probably increase the accuracy of REM staging when the dataset collected is not exhaustive, as demonstrated by some authors [89,90,91,92]. However, it is essential to consider that these studies are still at a preliminary stage.

In summary, PSG is the gold standard and EEG is the biopotential validated by doctors for sleep analysis. On the other hand, CST-mounted PPG and accelerometers are the most popular options, able to distinguish the sleep/wake phases, but unable to identify the REM stage and not clinically accepted. To improve the accuracy of REM staging by wearable technology, it would be desired to consider other markers in addition to HR/HRV and movement, without worsening the cost, comfortability, and wearability in comparison to CSTs.

In this scenario, we propose a wearable device using three QVAR sensors designed for recording three biopotentials: the EOG, the surface EMG (sEMG), and the ECG. The EOG, used to assess rapid eye movements, can be accurately captured from the Fp1 and Fp2 derivations of the International 10–20 System Standard. To assess muscle atonia, the sEMG can be taken from the masseter or mentalis muscles. The ECG, essential to derive HR and HRV, can be recorded using any derivation. The proposed device follows the strategy of comfort adopted by CSTs, while adding value through the inclusion of EOG and sEMG, at no additional cost. Leveraging the MEMS technology, the QVAR sensors are embedded in accelerometers and are available on the market at the same price. Furthermore, the minimal power consumption of a QVAR sensor during recording (27 µW) ensures that the energy expenditure is negligible, thus presenting a cost-effective solution compared to inertial acquisitions.

Our system is contained in an elastic headband, as illustrated in Figure 11. This headband includes a Mini Nucleo L031K6 board by STMicroelectronics, which embeds a Cortex M0 microcontroller, along with three QVAR sensors for acquiring the EMG, ECG, and EOG.

The electrodes on the masseter (M1, M2) for the EMG and the electrodes on the chest (RA, LA) for the ECG are shown in the photograph by the side. The electrodes on the frontal regions (Fp1, Fp2) for the EOG are inside the headband, in contact with the forehead skin. The ECG provides a comprehensive overview of the heart rate, which could also be detected through a PPG sensor in order to reduce wiring, despite increased power consumption. The QVAR sensors transmit data to the Nucleo through the I2C standard. Each QVAR acquires the corresponding biopotential at a sampling frequency of 240 Hz, utilizing a two-electrode configuration. The reference voltage for QVAR sensors is provided by the on-board circuitry, avoiding the third reference electrode.

Tests were performed on a control subject over consecutive nights. A result of main interest is reported in Figure 12. This figure represents a specific 45-min timeframe during a night, which started approximately 3 h after the subject fell asleep. The graphical representation reports the traces of the three biopotentials as a function of time, covering the interval from 3:10 to 3:55 a.m. (in these figures, the time scale has an internal origin). When considering the HR derived from the trace recorded by QVAR 1, initially it remained below 55 bpm in the first minutes; then, at 3:28, the HR exhibited wide variability, with bursts reaching up to 80 bpm. This interval of pronounced HRV stopped at 3:45, after which the HR reverted to values below 55 bpm. The 17-min interval between 3:28 and 3:45 is highlighted with a grey shadow. Concerning the eye activity detected by QVAR 2, the EOG signal remained close to zero for the initial 8 min, indicating an absence of eye movements. Then, around 3:28, there was a substantial increase in eye activity, which stopped at 3:45. Lastly, regarding the masseter tone detected by QVAR 3, in the shadowed interval, the masseter manifested total atonia, while in other intervals a residual constant contraction was present. The simultaneous occurrence of all three patterns was an extraordinary event, observable only two times throughout the night of testing, with durations of 17 and 15 min, respectively. On the other hand, in other intervals of variable durations, frequent changes in the individual signals under consideration occurred. In particular, QVAR 1 often recorded pronounced variability in HR, underscoring the limitations of CST in accurately analyzing sleep patterns.

## 5. Conclusions

We have introduced an innovative methodology for biopotential acquisition based on a commercial MEMS multisensor integrated platform mounted with electrostatic charge variation (QVAR) sensors. The main advantages of the QVAR sensor are its extremely low power consumption and its versatility in managing signals of different natures. Indeed, the QVAR sensor was originally intended for presence tracking in the automotive industry, so the existing setup has been engineered for the acquisition of biopotential, designing a dedicated front-end and writing proper firmware. The other MEMS features, as mentioned previously, can enable further useful functionalities without augmenting the cost, power consumption, and space requirements. For example, with an accelerometer on the nose, it is possible to detect the breathing rate during sleep and the motion due to obstructive apnea or involuntary movements [72]. To the best of our knowledge, this is the first time that an electrostatic sensor based on electric charge transfer has been used for health applications.

Preliminary tests were systematically performed to verify the correct acquisition of biopotentials. To this aim, ECG, EEG, EOG, and sEMG traces were recorded on control subjects using QVARs and gold standards simultaneously, under the supervision of doctors. The comparison was excellent in all of the tests, with no appreciable discrepancies between the two setups.

As applications of the QVAR technology for health purposes, we proposed two wearable systems, specifically for nocturnal use in domestic environments. One system records ECGs and one-channel EEGs for monitoring specific vital signs of diabetic patients before and during nocturnal hypoglycemic crisis. The other system records the EOG, ECG, and sEMG on the masseter muscle for screening the rapid eye movement (REM) sleep phase. In both cases, the nocturnal polygraph recording was successful and the subjects did not report any discomfort. This study was approved by the ethics committee. The participants of the study included a subset of subjects from the approved protocol, and they signed an informed consent form.

Clinical considerations are immature and beyond the scope of this paper.

As a conclusion, we can definitely affirm that MEMS QVAR technology provides an incredibly low-power, unexplored solution to biopotential acquisition, and it also enables this type of functionality to be added to any existing MEMS board at near-zero additional power consumption. For these reasons, those sensors open up additional possibilities for wearable sensing and strengthen the role of MEMS technology in medical wearable applications for the long-term synchronous acquisition of biopotential and other physical parameters.

## Figures and Tables

**Figure 1 sensors-24-01554-f001:**
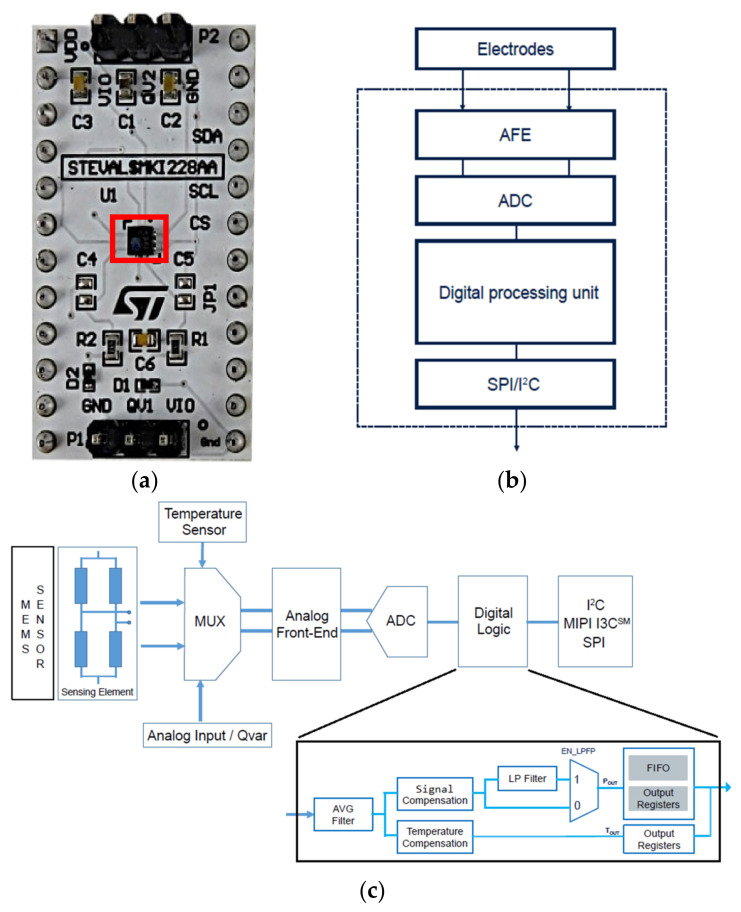
(**a**) QVAR evaluation board details: the sensor is highlighted by the red box. (**b**) High-level QVAR block diagram and electrodes. (**c**) Low-level functional block diagram of the QVAR sensor.

**Figure 2 sensors-24-01554-f002:**
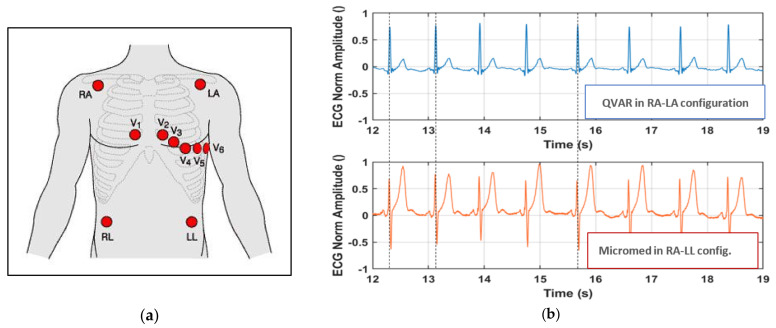
(**a**) Electrode placement for a 12-lead ECG configuration, with electrodes on the right arm (RA), left arm (LA), left leg (LL), right leg (RL), and chest (electrodes V1 to V6). (**b**) Normalized ECG traces acquired simultaneously by the QVAR and Micromed systems, from RA-LA and RA-LL configurations, respectively. Dashed lines highlight the R peaks.

**Figure 3 sensors-24-01554-f003:**
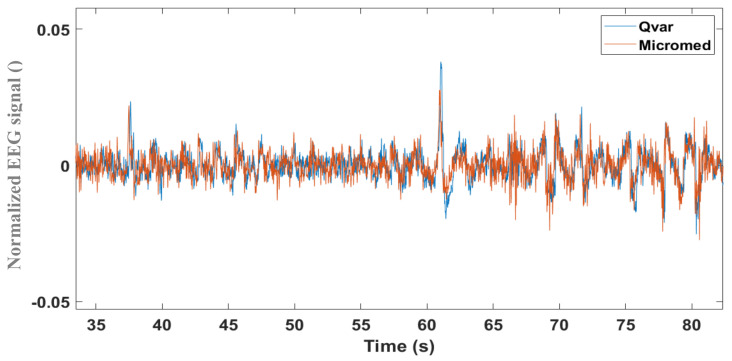
Raw traces of single-channel EEG acquisition from the frontal lobes by Micromed and QVAR sensors.

**Figure 4 sensors-24-01554-f004:**
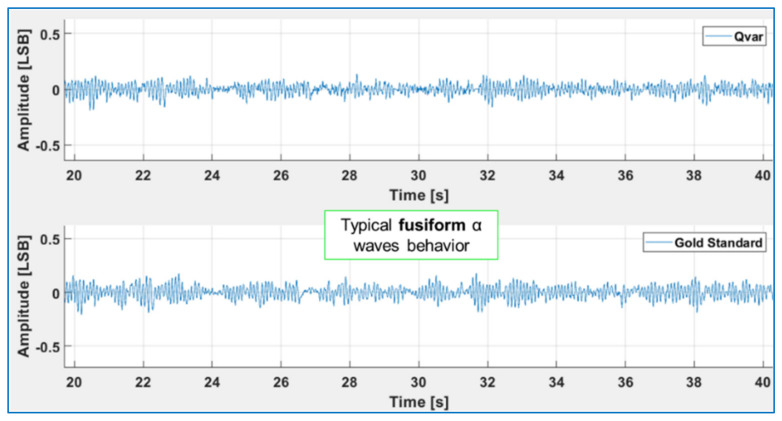
The 8–13 Hz filtered normalized signal acquired in O1-O2 with QVAR (**top**) and the gold standard (**bottom**), relative to the acquisition of an α-wave EEG with closed eyes.

**Figure 5 sensors-24-01554-f005:**
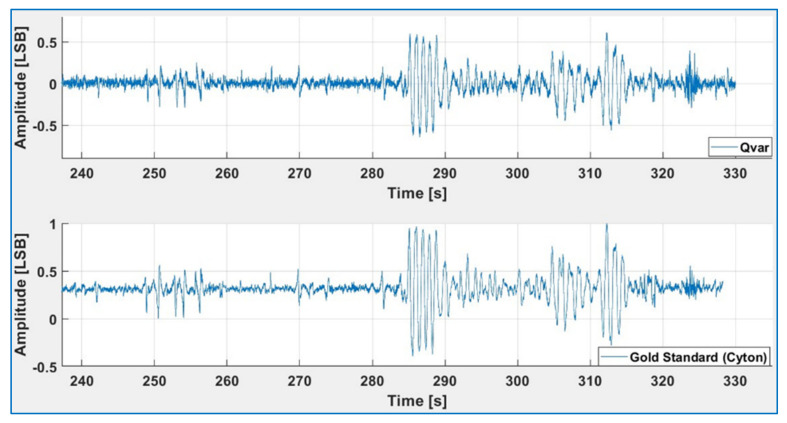
The 1–3 Hz filtered normalized signal acquired in FP1-FP2 with QVAR (**top**) and the gold standard (**bottom**), relative to the acquisition of a β-wave EEG with closed eyes and random mixed eye actions.

**Figure 6 sensors-24-01554-f006:**
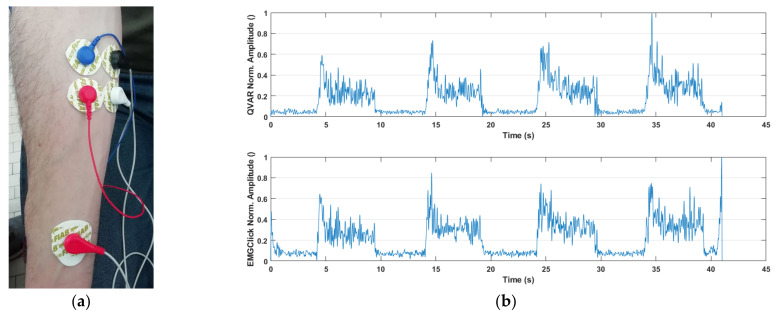
Picture of the patch electrodes positioned on the flexor carpi radialis (**a**). The traces collected by the QVAR (**top**) and EMGClick (**bottom**) devices are displayed by the side (**b**).

**Figure 7 sensors-24-01554-f007:**
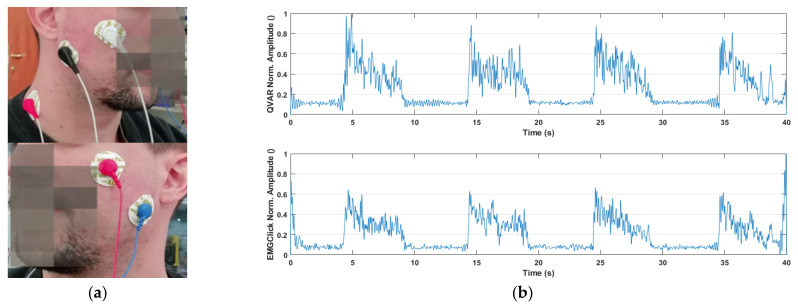
(**a**) Picture of the patch electrodes positioned on the masseter. (**b**) The traces collected by the QVAR (**top**) and EMGClick (**bottom**) devices are displayed by the side.

**Figure 8 sensors-24-01554-f008:**
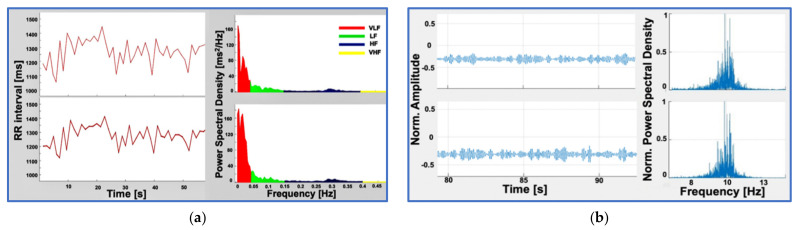
(**a**) RR trait duration and power spectral density obtained with QVAR (**top**) and the gold standard (**bottom**); (**b**) α-wave EEG and normalized power spectral density recorded by QVAR (**top**) and the gold standard (**bottom**).

**Figure 9 sensors-24-01554-f009:**
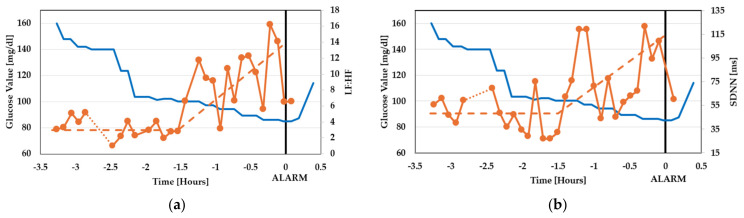
Values of the ECG parameters ((**a**) LF:HF, (**b**) SDNN) before the GM alarm at t = 0 (orange curve with symbols). In the same figure, the glucose data recorded by the GM are also drawn as solid blue lines. The dashed lines indicate the trend of the studied parameters.

**Figure 10 sensors-24-01554-f010:**
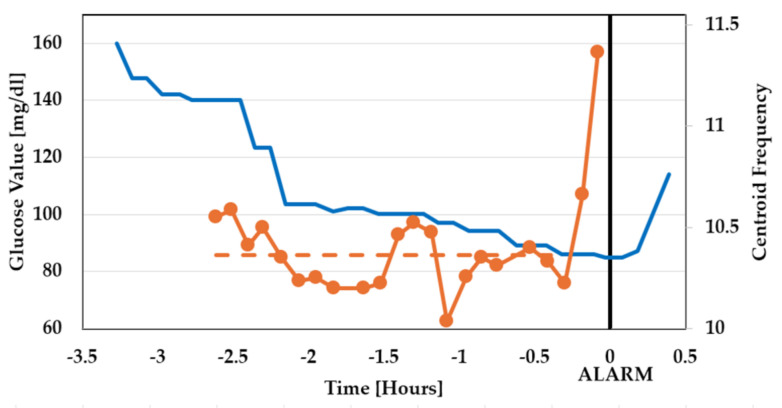
Values of the EEG parameter (CF) before the alarm at t = 0 (orange curve with symbols). In the same figure, the glucose data recorded by the GM are also drawn as a solid blue line. The dashed line indicates the trend of the studied parameter.

**Figure 11 sensors-24-01554-f011:**
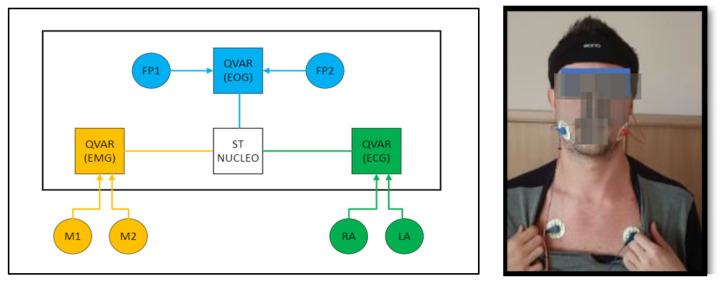
Sketch of the headband for REM sleep analysis, which includes an ST Nucleo board and three QVARs for the EMG, ECG, and EOG. The electrodes for the EMG (M1, M2) and ECG (RA, LA) come out of the band and are positioned as shown in the photograph. Electrodes for EOG (Fp1, Fp2) are embedded in the headband and lie in contact with the forehead skin.

**Figure 12 sensors-24-01554-f012:**
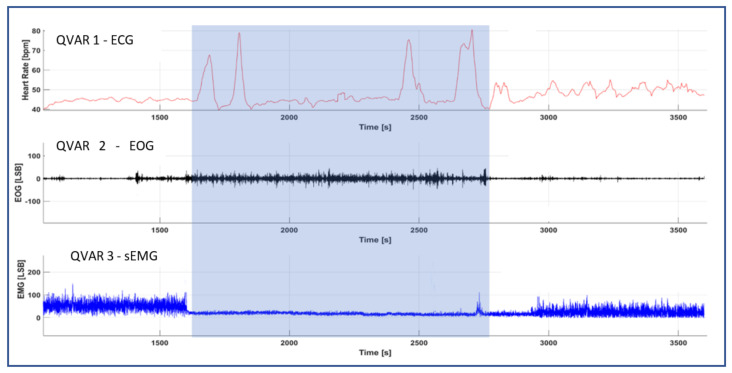
Traces of HR, EOG, and sEMG recorded by the three QVARs in a time interval of 45 min during the nocturnal test. The 17 min shadowed interval denotes the occurrence of simultaneous alterations of the three biopotentials.

**Table 1 sensors-24-01554-t001:** Typical values of the principal electrical parameters of QVAR at VDD = 1.8 V and 25 °C.

Electrical Parameters @ VDD = 1.8 V, T = 25 °C	Typ.
Supply voltage	1.71 V to 3.6 V
I/O pins supply voltage	1.08 V to 3.6 V
Current consumption	190 uA
Current consumption in power-down mode	2.6 uA
Digital high-level input voltage	0.7 × VDDIO
Digital low-level input voltage	0.3 × VDDIO
Digital high-level output voltage	VDDIO—0.2 V
Digital low-level output voltage	0.2 V

**Table 2 sensors-24-01554-t002:** Typical values of the principal electrical characteristics of QVAR at VDD = 1.8 V and 25 °C.

Electrical Characteristics @ VDD = 1.8 V, T = 25 °C	Typ.
ODR (Configurable output data rate)	120 to 240 Hz
Input range (DC-coupled)	±460 mV
Offset (input referred)	±3 mV
Noise (shorted input)	54 uV_RMS_
QVAR gain	78 LSB/mV
CMRR	54 dB
Input impedance (configurable)	235 to 2400 MΩ

**Table 3 sensors-24-01554-t003:** Values of the main ECG/EEG parameters averaged over 24-h acquisition on six control subjects.

ECG/EEG Parameters	QVAR System	Gold Standard	Discrepancy %
QTc (ms)	329	334	1.4
RR (ms)	948.4	957.6	0.95
SDNN	149.9	149.1	0.5
LF:HF	0.595	0.590	0.84
CF (Hz)	102.13	102.28	0.1

## Data Availability

Data are contained within the article.

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
