# Peer review of "Multisensor Integrated Platform Based on MEMS Charge Variation Sensing Technology for Biopotential Acquisition"

_sensors, 2024, doi:10.3390/s24051554_

Round 1
Reviewer 1 Report
Comments and Suggestions for Authors
The paper presents a MEMS multi-sensor integrated system designed for long-time recording of biopotentials and inertial data. It effectively describes the architecture of the sensor and provides comprehensive electrical specifications. Additionally, the sensor's performance is evaluated through real data measurements, and the results are validated.
Overall, the subject matter is compelling, and the paper is suitable for publication. However, I recommend the following minor modifications:
-
Improvement of Figure Quality: The quality of the figures could be enhanced. In some instances, it is challenging to discern the data, particularly when zoomed in. Upgrading the resolution or employing clearer visualization techniques would enhance readability.
-
Modification of Paper Structure: Consider restructuring the paper to consolidate the presentation of results. Currently, the outputs are dispersed across sections 3 and 4. Consolidating them into a single "Results" section might improve clarity, especially for papers with a substantial number of findings.
Author Response
Dear Reviewer,
we thank you for your consideration and appreciation. We tried to accomplish your suggestions, re-making some figures and tables with an improved resolution and re-arranging the structure of the paper including all the new results in a single section. All the changes are listed in the attached letter and are highlighted along the revised version of the manuscript.
Reviewer 2 Report
Comments and Suggestions for Authors
This paper makes an excellent attempt and verifies the test results. The results show that it is feasible for QVAR sensor to be used in biological testing. But there are some minor problems that need to be considered.
1 The test results in figure 3 are quite different at the position of the T wave. Why? Will such test results affect the doctor's diagnosis?
2 QVAR sensors are mainly used in automobile system, does it need to be modified in order to be suitable for the wearable system? I think the author needs to explain.
3 On line 3 on page 14, the reference is lost.
Author Response
Dear Reviewer,
we thank you for your consideration and appreciation. We tried to answer to your questions. The co-authors doctors answered to your question #1. All the changes are listed in the attached letter and are highlighted along the revised version of the manuscript.

Reviewer 3 Report
Comments and Suggestions for Authors
Author Response
Dear Reviewer,
we thank you for your consideration and appreciation. We tried to answer to your questions and accomplish suggestions. Co-authors from STMicroelectronics disclosed the information required in your question #1, and comments and a new figure have been added. The introduction has been improved in order to accomplish your suggestion #3 . All the changes are listed in the attached letter and are highlighted along the revised version of the manuscript.

Round 2
Reviewer 3 Report
Comments and Suggestions for Authors
The manuscript looks perfect after revisions. Thank you.
A typo was found in the first paragraph of 4.4 "for the acquisition of EO?"
Also, looks like it is a pure function of ASIC in the package for the measurement but MEMS itself is not involved. I am curious why STMicroelectronics does not provide a separate IC product for the sensing.
Thank you.
Author Response
We thank again the Reviewer and corrected the mistake.
Also write an answer to her/his curiosity in the point-by-point letter
